**Data Availability Statement:** All relevant data are within the manuscript and its Supporting Information files.

## RESEARCH ARTICLE

# *Plasmodium falciparum* serology: A comparison of two protein production methods for analysis of antibody responses by protein microarray

Tate Oulton[1], Joshua Obiero[2], Isabel Rodriguez[3], Isaac Ssewanyana[4,5], Rebecca A. Dabbs[1], Christine M. Bachman[6], Bryan Greenhouse[3], Chris Drakeley[1], Phil L. Felgner[2], Will Stone[1], Kevin K. A. Tetteh[1]*

1 Department of Infection Biology, London School of Hygiene and Tropical Medicine, London, United Kingdom, 2 Department of Physiology and Biophysics, University of California, Irvine, Irvine, CA, United States of America, 3 Department of Medicine, University of California San Francisco, San Francisco, California, United States of America, 4 Infectious Diseases Research Collaboration, Kampala, Uganda, 5 Makerere University College of Health Sciences, Kampala, Uganda, 6 Global Health Labs, Inc, Bellevue, Washington, United States of America

* kevin.tetteh@lshtm.ac.uk

## Abstract

The evaluation of protein antigens as putative serologic biomarkers of infection has increasingly shifted to high-throughput, multiplex approaches such as the protein microarray. In vitro transcription/translation (IVTT) systems–a similarly high-throughput protein expression method–are already widely utilised in the production of protein microarrays, though purified recombinant proteins derived from more traditional whole cell based expression systems also play an important role in biomarker characterisation. Here we have performed a side-by-side comparison of antigen-matched protein targets from an IVTT and purified recombinant system, on the same protein microarray. The magnitude and range of antibody responses to purified recombinants was found to be greater than that of IVTT proteins, and responses between targets from different expression systems did not clearly correlate. However, responses between amino acid sequence-matched targets from each expression system were more closely correlated. Despite the lack of a clear correlation between antigen-matched targets produced in each expression system, our data indicate that protein microarrays produced using either method can be used confidently, in a context dependent manner, though care should be taken when comparing data derived from contrasting approaches.

## Introduction

To date, the majority of malaria serologic studies have focussed on antibody responses to a small number of well-characterised, highly immunogenic *Plasmodium falciparum* antigens that have proven to be reliable markers of exposure to infection [1–8]. However, *P. falciparum*

**Funding:** KKAT was supported by a Bloomsbury SET Award (Innovation Fellowship to KKAT; BSA14) under the UKRI Connecting Capabilities Fund (CCF). WS is supported by a Sir Henry Wellcome fellowship (number 218676/Z/19/Z) from the Wellcome Trust (UK). The study was also supported by an award from the Global Good Fund I, LLC awarded to CD and KKAT. The funders had no role in study design, data collection and analysis, decision to publish, or preparation of the manuscript.

**Competing interests:** The authors have declared that no competing interests exist.

**Abbreviations:** IVTT, in vitro Transcription/ Translation.

expresses more than 5000 proteins, each a potential antibody target [9, 10]. Advances in technology have led to the development of new assay platforms that allow proteome scale investigation of antibody responses, such as the protein microarray [11, 12]—boasting significantly greater experimental throughput than more classical monoplex methods (e.g. ELISA) [13, 14]. The ability to simultaneously interrogate large numbers of putative targets, using low volumes of sample, significantly increases the rate at which an individual's antibody responses to antigens can be characterised. As such, protein microarray based approaches to biomarker identification and humoral response profiling in malaria, and other infectious diseases, have been increasingly adopted [15–24].

One widely utilised form of the protein microarray is based on an in vitro transcription/ translation (IVTT) system [25]–where protein products are produced through a PCR, in vivo recombination cloning and an in vitro expression pipeline, before being printed onto arrays [15]. In principle, whole organism proteome microarrays can be fabricated simply and quickly, enabling analysis of all potential protein driven immune responses to a pathogen. Cell-free synthesis (CFS) is a technique first established over 50 years ago as a means to dissect the molecular mechanisms around protein expression. More recently, the technique has been used as a high throughput expression platform to explore a number of diverse biological processes [26, 27]. At its simplest, the approach utilises the crude extract containing the transcription and translation machinery from the cell, performing the process of protein expression without the constraints of the cell. This allows a wide variety of proteins to be expressed including those that would be deemed toxic if expression was attempted within the confines of the cell membrane [28]. CFS systems based on *Escherichia coli* (*E.coli*) are among the most widely used of the IVTT systems [27] and have helped to transform the narrative around a number of areas including biomarker discovery for infectious diseases [15, 29, 30]. Despite the widespread uptake of the approach there remain some issues around the technique. This includes significant heterogeneity of expression, leading some research groups to describe the mechanisms of the process as a "black box". Therefore, the inherent heterogeneity between products is not assessed for every target making it difficult to normalise for reactivity between protein spots, which represent an impure mix of *E. coli* and target protein. In addition to the *E. coli* cell-free expression platform, other approaches have been employed in the characterisation of protein targets for immunological assessment. The wheat germ cell-free expression system in particular has also proven to be an important platform in the advancement of biomarker discovery and malaria vaccine research [31–34]. This is not the focus of the current study.

In contrast to the IVTT array methodology, the printing of purified proteins is cheaper and typically more quantifiable. Uniform amounts of product can therefore be incorporated into arrays, increasing confidence when comparing quantitative antibody responses between antigenic targets [35] and assessing relative immunogenicity. The process can be modified to support the scale up of recombinant proteins, and furthermore, affinity purification of protein targets reduces the risk of undesired background reactivity due to expression system components, and in part truncated proteins. However, the time required to produce panels of purified proteins is far in excess of the IVTT system, particularly for large numbers of targets, unless supported by an automated production platform [36–38]. For both the IVTT and purified protein *E. coli* systems, although the production of complex conformational proteins is possible it can sometimes be a challenge [39, 40]. These challenges are in part due to the expression of proteins foreign to the bacteria, the speed at which bacteria express proteins, only partially mitigated with a reduction in expression temperature; and the lack of essential molecular chaperones to aid correct folding/refolding of proteins [41–43].

Here we present a comparison between IVTT based and purified proteins on a single microarray. For clarity proteins produced using the IVTT system will simply be referred to as

IVTT proteins, and those produced by conventional *E.coli* expression will be referred to as purified proteins. Matched malarial protein targets from each methodology were assessed for comparative reactivity in serum from Ugandan participant samples (n = 899) [44] to determine the suitability of each approach in the context of high-throughput profiling of serological responses to protein antigens.

## Material and methods

### Ethics statement

All serum samples were collected after written informed consent from the participant or their parent/guardian. The protocol for sample collection was reviewed and approved by the Makerere University School of Medicine Research and Ethics Committee (#2011–149 and #2011–167), the London School of Hygiene and Tropical Medicine Ethics Committee (#5943 and #5944), the Durham University School of Biological and Biomedical Sciences Ethics Committee, the University of California, San Francisco, Committee on Human Research (#11–05539 and #11–05995) and the Uganda National Council for Science and Technology (#HS-978 and #HS-1019).

### Samples

Sera were originally collected as part of a comprehensive longitudinal surveillance study conducted in three sub-counties in Uganda (Walukuba, Jinja District; Kihihi, Kanungu District, and Nagongera, Tororo). The study design and methods have been previously reported and are described in detail elsewhere [44]. A sub-selection of samples (n = 899) was made from individuals across a breadth of recorded clinical episodes of malaria to ensure a range of seroreactivity.

### Protein targets

**Purified protein expression.** Recombinant proteins were generated and expressed in *Escherichia coli* as glutathione S-transferase (GST)-tagged fusion proteins using previously described methods: PfMSP1-19 [45]; MSP1 block 2 [46]; ACS5, ETRAMP4 & HSP40 [19]; ETRAMP5 [19, 47]; EBA181 [48]; MSP4 [49]; MSP5 [50]; MSP7 [51]; and GAMA [52]. The exception to this was PfAMA1, which was expressed as a histidine tagged protein in *Pichia pastoris* [53]. Purification of the expressed proteins was performed using affinity chromatography (Glutathione Sepharose 4B (GE Healthcare Life Sciences) or HisPur Ni-NTA (Invitrogen) resins for GST and His tagged proteins, respectively). Protein concentration was assessed using the Bradford protein assay, with quality, and purity assessed by resolution on a 4–20% gradient SDS-PAGE.

**IVTT protein expression.** An IVTT system was used to express proteins of interest as previously described [15]. Briefly, *Plasmodium falciparum* DNA (3D7 isolate) coding sequences were PCR-amplified and cloned into T7 expression vectors via homologous recombination. Target sequences were expressed at 21˚C for 16h in *E. coli*-based, cell-free transcription/translation reactions, and products were printed onto arrays as un-purified, whole reaction mixtures.

**Overview of compared IVTT and purified protein antigens.** We assessed antibody responses to protein targets mapping to eleven antigens (i.e. distinct gene products), each represented on the array by at least one IVTT and one purified protein target. Full details are in **Table 1** and **S1 Table**. The number of purified protein targets varied according to availability, while the number of IVTT targets was dependent on the exon composition of each the gene

**Table 1. Description of *P.falciparum* antigens and their corresponding IVTT and purified protein targets.**

| Protein | Description | Full length (amino acids) | Protein target/expression system | Size (Start amino acid—End amino acid) |
|---|---|---|---|---|
| ACS5 | Acyl CoA synthase | 811 | IVTT_1 | 811 (1–811) |
| | | | Pure_1 | 117 (294–410) |
| | | | Pure_2 | 160 (414–573) |
| | | | Pure_3 | 150 (578–727) |
| AMA1 | Apical membrane antigen 1 | 622 | IVTT_1 | 622 (1–622) |
| | | | Pure_1 | 450 (97–546) |
| EBA181 | Erythrocyte binding antigen 181 | 1567 | IVTT_1 | 754 (1–754) |
| | | | IVTT_2 | 752 (737–1488) |
| | | | Pure_1 | 585 (755–1339) |
| ETRAMP4 | Early transcribed membrane antigen 4 | 136 | IVTT_1 | 136 (1–136) |
| | | | Pure_1 | 25 (28–52) |
| | | | Pure_2 | 61 (76–136) |
| ETRAMP5 | Early transcribed membrane antigen 5 | 181 | IVTT_1 | 181 (1–181) |
| | | | Pure_1 | 86 (26–111) |
| | | | Pure_2 | 47 (135–181) |
| GAMA | GPI-anchored membrane antigen | 738 | IVTT_1 | 738 (1–738) |
| | | | Pure_1 | 99 (68–166) |
| HSP40 | Heat shock protein 40 type II | 402 | IVTT_1 | 134 (80–213) |
| | | | IVTT_2 | 171 (213–401) |
| | | | Pure_1 | 83 (71–153) |
| | | | Pure_2 | 189 (214–402) |
| MSP1 | Merozoite surface protein 1 | 1720 | IVTT_1 | 870 (1–870) |
| | | | IVTT_2 | 868 (853–1720) |
| | | | Pure_1 | 45 (64–108) |
| | | | Pure_2 | 35 (54–63;109–133) |
| | | | Pure_3 | 116 (1605–1720) |
| MSP4 | Merozoite surface protein 4 | 272 | IVTT_1 | 162 (1–162) |
| | | | IVTT_2 | 161 (1–161) |
| | | | IVTT_3 | 110 (163–272) |
| | | | Pure_1 | 65 (43–107) |
| MSP5 | Merozoite surface protein 5 | 272 | IVTT_1 | 172 (1–172) |
| | | | IVTT_2 | 171 (1–171) |
| | | | Pure_1 | 61 (147–207) |
| MSP7 | Merozoite surface protein 7 | 351 | IVTT_1 | 351 (1–351) |
| | | | Pure_1 | 175 (177–351) |

sequence; multiple exon sequences were expressed as multiple protein targets based on exon delineation. Similarly, single exon gene sequences were generally expressed as a single protein. As a result, of the 11 antigens investigated, 8 were represented by >1 IVTT or purified protein target; 5 had >1 IVTT protein target (EBA181, HSP40, MSP1, MSP4 and MSP5) and 5 had >1 purified protein target (ACS5, ETRAMP4, ETRAMP5, HSP40 and MSP1). Near identical IVTT proteins (1 terminal amino acid difference in length) were produced independently and printed in parallel for two antigens: MSP4 and MSP5 as expression controls. Sequence information used in the design and expression of the purified *E.coli* proteins were generally smaller than the equivalent proteins expressed in the IVTT cell-free systems. This was done to limit the sequence length to below 1kb as expression of proteins larger that 1kb in *E.coli* can contribute to poor or failed expression yields [42, 43]. Truncation of target sequences was based on in

silico mapping of each protein sequence to focus on regions of predicted immunogenicity based on the in silico analysis. Empty GST vectors were expressed and the purified GST used in background correction for proteins with this tag. His-tag vector was not expressed as it has proven impossible to express and purify the 6xhistidine tag in isolation.

## Protein microarray

Prior to printing, Tween 20 was added to purified proteins to yield a final concentration of 0.001% Tween 20. Arrays were printed onto nitrocellulose-coated slides (AVID, Grace Bio-Labs, Inc., Bend, OR, USA) using an Omni Grid Accent microarray printer (Digilabs, Inc., Marlborough, MA, USA). Alongside proteins of interest, buffer (PBS) and no-DNA (empty T7 vector reactions) were included as controls to allow for background normalisation of purified and IVTT proteins respectively.

**Sample probing.**   For analysis of antibody reactivity on the protein microarray, serum samples were diluted 1:200 in a 3 mg mL$^{-1}$ *E. coli* lysate solution in protein arraying buffer (Maine Manufacturing, Sanford, ME, USA) and incubated at room temperature for 30 min. Arrays were rehydrated in blocking buffer for 30 min. Blocking buffer was removed, and arrays were probed with pre-incubated serum samples using sealed, fitted slide chambers to ensure no cross-contamination of sample between pads. Chips were incubated overnight at 4˚C with agitation. Arrays were washed five times with TBS-0.05% Tween 20, followed by incubation with biotin-conjugated goat anti-human IgG (Jackson ImmunoResearch, West Grove, PA, USA) diluted 1:200 in blocking buffer at room temperature. Arrays were washed three times with TBS-0.05% Tween 20, followed by incubation with streptavidin-conjugated SureLight P-3 (Columbia Biosciences, Frederick, MD, USA) at room temperature protected from light. Arrays were washed three times with TBS-0.05% Tween 20, three times with TBS, and once with water. Arrays were air dried by centrifugation at 500 x *g* for 5 min and scanned on a GenePix 4300A High-Resolution Microarray Scanner (Molecular Devices, Sunnyvale, CA, USA). Target and background intensities were measured using an annotated grid file (. GAL).

**Data normalisation.**   Microarray spot foreground and local background fluorescence data were imported into R (Foundation for Statistical Computing, Vienna, Austria) for correction, normalisation and analysis. Local background intensities were subtracted from foreground using the backgroundCorrect function of the limma package [54]. The backgroundCorrect function was then further applied to GST-tagged purified proteins, whereby background-corrected GST fluorescence was subtracted from background-corrected target fluorescence to account for any GST-specific reactivity in samples. All data were then Log2 transformed and the mean signal intensity of buffer and no-DNA control spots were subtracted from purified and IVTT proteins respectively to give a relative measure of reactivity to targets over background (**S1 Fig**) [20].

## Results

**Table 1** summarises the purified and IVTT protein targets for each antigen, with further detail in **S1 Table**. In brief, we assessed IgG antibody responses to 35 antigenic targets, derived from 11 well-characterised *P. falciparum* protein antigens (distinct gene products). Each antigen was represented by at least one IVTT and one purified protein target.

### Magnitude of responses between expression systems

The magnitude of response to all protein targets was compared by antigen to evaluate differences in seroreactivity between IVTT derived and purified protein targets. As expected,

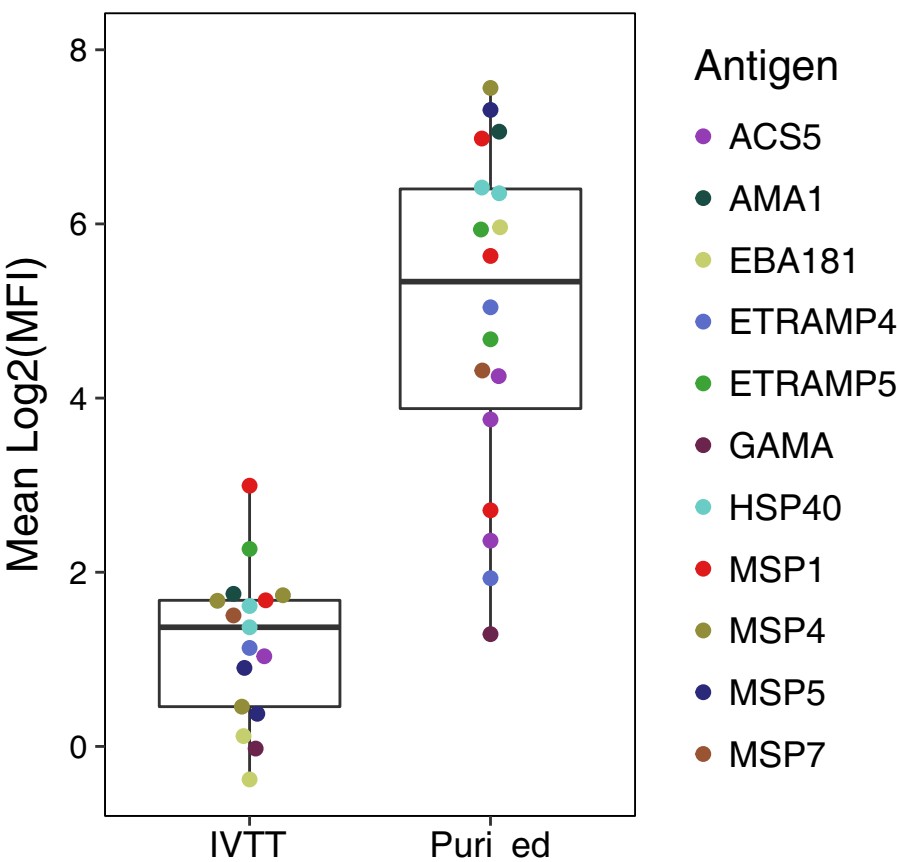

**Fig 1. Mean magnitude of antibody responses to targets.** The mean magnitude of response of each protein target stratified by expression system, presented with median and interquartile range of all mean responses.

responses varied significantly between antigens and between the protein targets mapping to each antigen.

Mean responses to all targets were compared by expression system (**Fig 1**) revealing a greater range of response to purified proteins (IQR Log2MFI = 3.88–6.40) than IVTT proteins (IQR Log2MFI 0.46–1.68), and a greater magnitude of response to purified than IVTT targets (p = <0.001). Similarly, the range and median intensity of individual antibody responses was found to be greater for purified proteins than their IVTT counterparts (e.g. AMA1—IVTT_1, median [IQR] Log2MFI = 1.66 [0.80–2.53]; Pure_1, median [IQR] Log2MFI = 7.92 [6.16–8.52]) for all targets (p = <0.001) except MSP1 Pure_2, which more closely reflected the level of reactivity to the two MSP1 IVTT targets (**Fig 2**).

## Correlation of responses between antigen matched targets

Considering all at least partially sequence matched IVTT and purified protein targets (i.e. excluding pairwise comparisons where purified protein sequence were completely non-overlapping with IVTT sequence for the same antigen) there was no evidence for a general correlation in mean response between expression platforms (Spearman's rho ($r_s$) = 0.279, p = 0.23). Antibody responses to all protein targets for each antigen were therefore compared

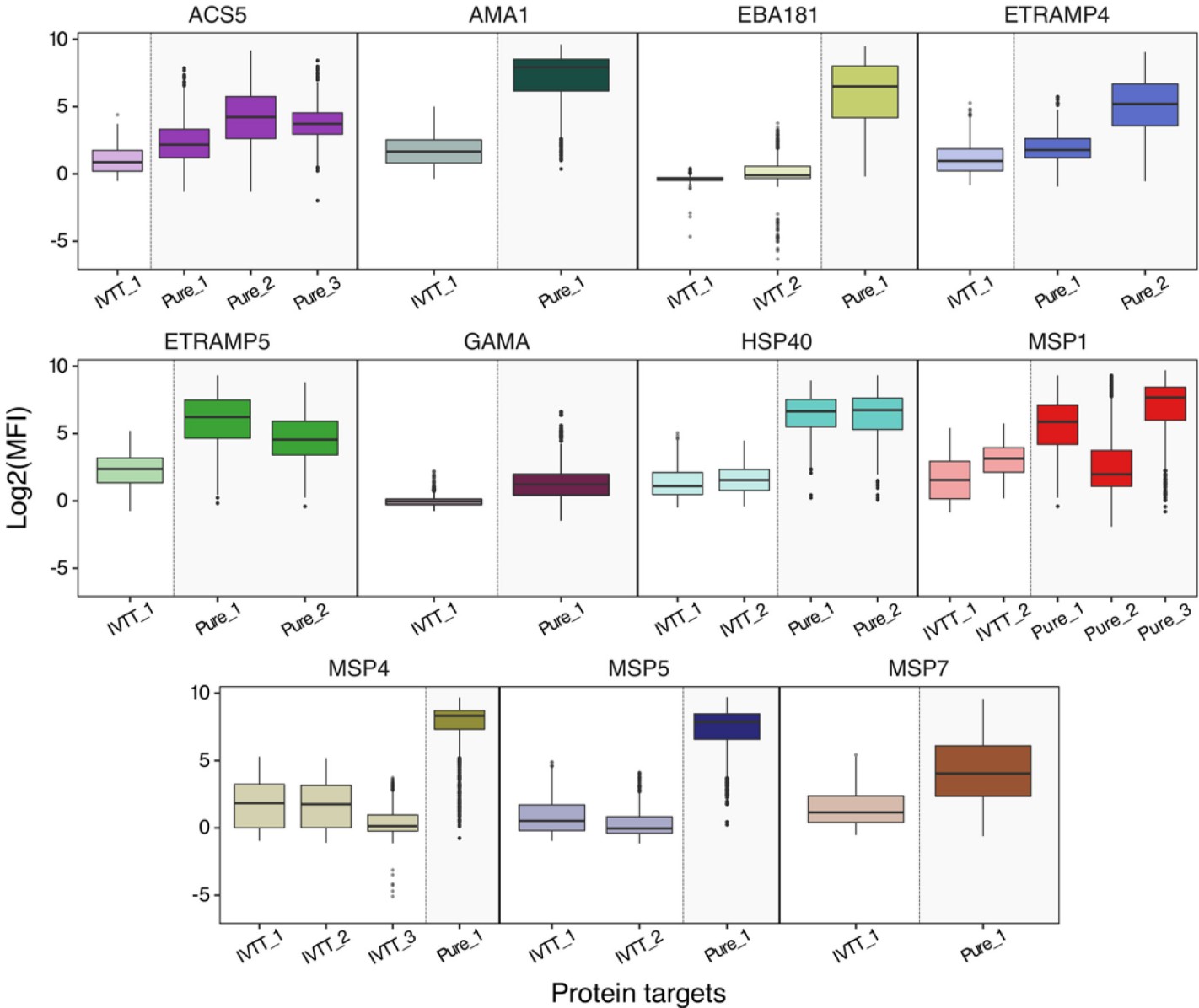

**Fig 2. Magnitude and range of response to IVTT and purified proteins.** All sample responses (n = 899) to all protein targets grouped by antigen, presented with median and interquartile range.

individually (representative example in **Fig 3** and all antigens in **S2 Fig**). This allowed for comparison between sequence matching IVTT and purified protein targets (e.g. HSP40 IVTT 2 vs. HSP40 Pure 2), non-matching IVTT and purified protein targets (e.g. HSP40 IVTT 2 vs. HSP40 Pure 1), and matching or non-matching targets produced in the same system (e.g. HSP40 IVTT 1 vs. HSP40 IVTT 2). Correlations were highly variable ($r_s$ = 1.00 to -0.045) though all but one (GAMA; $r_s$ = -0.045, p = 0.17) demonstrated a degree of positive, if not always statistically significant, association.

Multiple IVTT targets were produced for EBA181, HSP40, MSP1, MSP4 and MSP5. For all other than MSP5, non-sequence matching IVTTs were produced; correlation co-efficient for these targets were between 0.37 and 0.73 (**S2 Fig**). For EBA181 and MSP1, IVTT targets

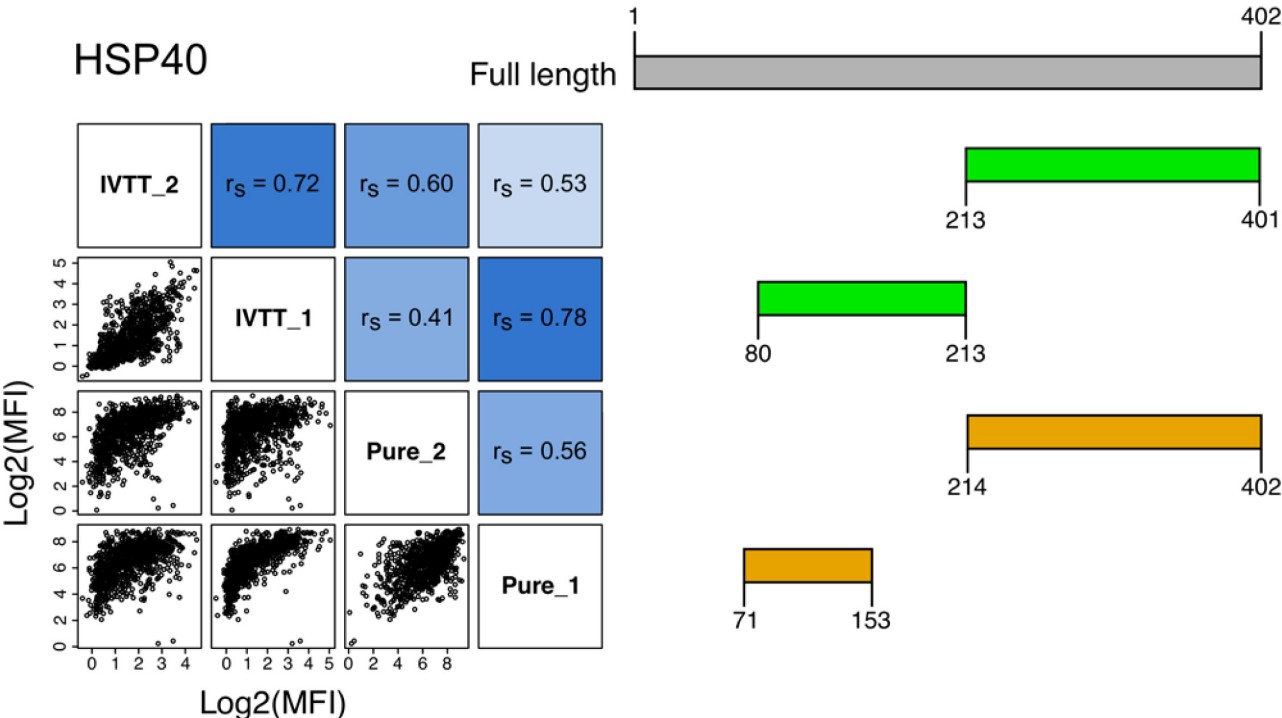

**Fig 3. Correlation of antibody responses and sequence mapping.** A representative example correlogram of multiple antigen-matched targets (left). Spearman's rank correlation reported ($r_s$) and increasing blue colour scale indicates relative strength of correlation based on calculated correlations for all proteins included in this analysis. Protein schematic (right) represents amino-acid aligned representation of IVTT (green) and purified (orange) proteins to the full-length native protein (grey). Proteins in the correlogram and schematic are correspondingly aligned. Corresponding axes are adjacently below or to the left of each protein.

overlap by 17 amino acids–equivalent to a small peptide in terminal regions unlikely to cover immunogenic epitopes. As such, these targets were considered non-overlapping. For MSP4 and MSP5, duplicate IVTT protein products were generated for each gene, with each duplicate protein identical to the other except for the omission of one [N- or C-] terminal amino acid. These respective targets resulted in near perfect correlation of antibody responses (MSP4 $r_s$ = 1.00, p = <0.001; MSP5 $r_s$ = 0.94, p = <0.001). Multiple purified protein targets were produced for ACS5, ETRAMP4, ETRAMP5, HSP40 and MSP1—none of which overlap. Correlation between these purified protein targets in each antigen varied between 0.31 and 0.59 (S2 Fig).

For the 8 antigens with >1 IVTT or purified protein target, the greatest level of correlation was found between an IVTT and purified target in 4/8 instances; between two IVTT targets (IVTT-IVTT) in 3/8 instances; and between two purified targets (purified-purified) in 1/8 instances (S2 Table). Comparing correlations between antigen-matched IVTT and purified proteins only, overlapping targets correlate more highly than non-overlapping targets. Sample sizes were too low to test the significance of this trend within antigens (Fig 4).

## Discussion

Protein microarrays are a practical approach to the serological screening of large numbers of putative malaria antigen biomarkers. The throughput and flexibility of the microarray platform presents an opportunity to interrogate malarial antibody responses at a scale far exceeding traditional mono- or multiplex approaches, agnostic of predicted immunological targets. Here we have evaluated matched antigenic targets produced using two *E. coli*-based expression

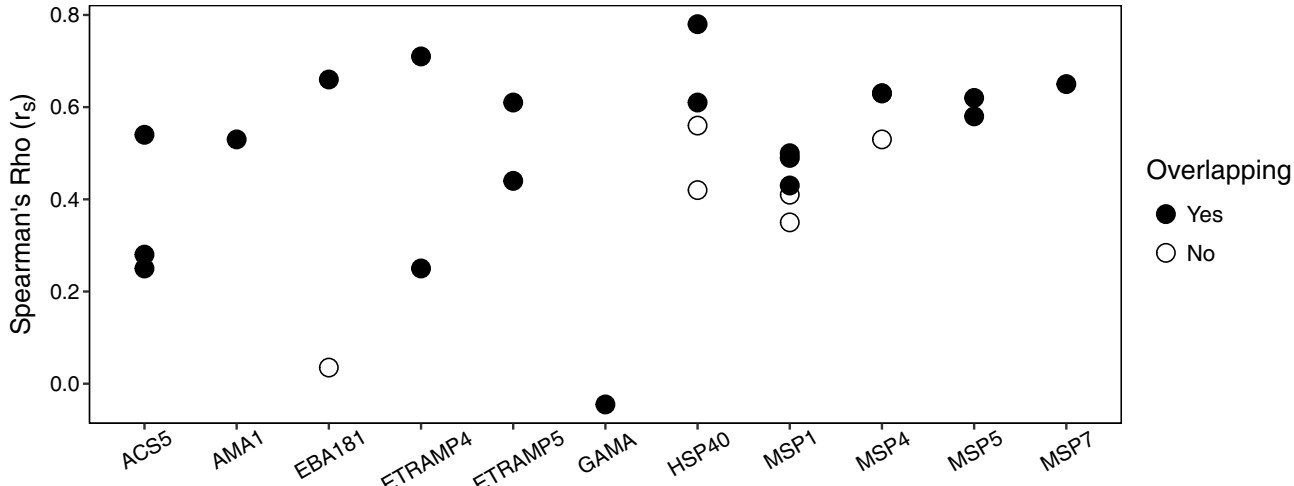

**Fig 4. Spearman's rho correlation coefficients between antigen-matched IVTT and purified proteins.** Targets with overlapping amino acid sequences are indicated by closed circles, compared to non-overlapping sequences indicated by open circles.

techniques–in vitro transcription/translation (IVTT), and purified, whole-cell recombinants–in the context of a protein microarray. We found that the magnitude of antibody responses to purified protein targets was generally higher than for their IVTT counterparts, and that correlation between protein target pairs at the individual serum sample level was variable and related to degree of sequence homogeneity between targets. Our findings warn against direct comparisons of microarray data from proteins produced in different expression platforms without careful cross-validation of sequences and allelic types. However, our data do provide support for the use of both IVTT and purified protein microarray platforms in the context of early-stage antigen biomarker identification to feed into experimental pipelines where candidate proteins may be interrogated by methods providing higher resolution analysis.

In building this study, we predicted that the magnitude of responses to IVTT products–which tended to be longer, often representing single exon sequences and therefore potentially containing more epitopes–would be greater than purified targets truncated based on species-specificity or domain boundaries which potentially represented fewer epitopes. Contrary to this prediction, we found that purified proteins captured a greater range and magnitude of responses (Purified, IQR Log2MFI = 3.88–6.40; IVTT, IQR Log2MFI 0.46–1.68; p = <0.001). The greater level of reactivity to purified targets may relate to differences in the amount of protein deposited on the array, where consistent and defined amounts of purified protein are spotted in contrast to the unquantified, and likely variable IVTT products. These findings recommend a degree of caution in interpretation of array data from two different platforms, for example: MSP5 showed the second highest mean MFI for any purified protein, but showed among the lowest mean MFI of any IVTT protein.

In addition to differences in the magnitude of mean responses to targets stratified by expression system, we observed a greater range of individual sample responses, stratified by antigen, to purified proteins than in sequence matched IVTT-expressed targets (e.g. AMA1—IVTT_1, median [IQR] Log2MFI = 1.66 [0.80–2.53]; Pure_1 (*Pichia pastoris* produced), median [IQR] Log2MFI = 7.92 [6.16–8.52]; p = <0.001). The *P. pastoris* AMA1 was included as a control for the evaluation of the production of a conformational protein. AMA1 is a complex structure comprised of three domains defined by three disulphide bonds. Production of AMA1 in *P. pastoris* has been fully characterised in terms of correct folding of the purified

protein [53, 55] and this observation is likely a reflection of antibody reactivity to correctly folded (*P. pastoris*) and incorrectly folded AMA1 (IVTT). We acknowledge that a lack of correct folding in other purified and IVTT products may impact on epitope recognition by antibodies raised to native protein during infection. However, human antibody responses are composed of a polyclonal response to each antigen, which will include both confirmation and linear epitopes. Whilst questions remain about the appropriateness of using unfolded protein fragments in serological screens, such reagents remain the most widely utilised and efficient approach in this context at present.

Considering all antigenic targets together, we found no evidence of correlation in mean reactivity to sequence matched targets between expression systems ($r_s$ = 0.28, p = 0.23). In the context of this study, this was not unexpected taking into account the differences observed in magnitude of response between IVTT and purified proteins, and that the length of native protein sequence coverage between IVTT and purified targets was highly variable. More broadly, it is perhaps less reassuring that matched targets derived from different expression systems lack more obvious relationships in antibody response than have been demonstrated in other studies [17, 56], though Kobayashi et al. report relatively similar results for a smaller number of targets expressed in *E. coli* (purified proteins) and IVTT systems specifically [30]. It is likely that protein concentration disparities between the two approaches are one of the drivers of this heterogeneity. However, without attempting to quantify the exact amount of protein generated in the small volume of IVTT reactions we are unable to address this here. Although in this current study targets grouped by antigen displayed highly variable correlations of response, it is encouraging that sequence matched proteins did generally display stronger correlations of response than non-sequence matched targets. Further, this may indicate the importance of capturing specific epitopes within expression sequences when producing antigens by either expression method.

Despite the lack of a clearly defined relationship between antigen-matched targets from the evaluated expression systems, we remain confident that microarrays utilising IVTT or purified recombinant proteins are able to produce compelling and biologically relevant data. Indeed, our data show age-dependent trends in antibody responses (typical of highly endemic populations) [1–3] irrespective of expression system (**S3 Fig**), lending weight to the applicability of either methodology in serological assays [57–70].

The IVTT system lends itself to microarray applications, as vast numbers of proteins, or even entire proteomes, may be produced at scale relatively quickly. However, for application to serology there is concern that expressed proteins are not quantified before printing, and that expression levels of product may vary considerably; product yield in bacterial-based IVTT systems is generally considered to be lower (typically ~1 mg mL$^{-1}$ or less) though higher protein yields have been reported [71, 72]. This has been shown to be due to an inherent heterogeneity with IVTT components, although this weakness is an area of active research [26, 73]. Similarly, it is important to acknowledge that the un-purified nature of printed reaction mixtures may mask, or otherwise adversely affect, the detection of antibody reactivity in a sample; Davies et al. report IVTT reaction compositions of 99% *E. coli* lysate to 1% target protein [15], though this will vary considerably, at scale, in practice.

In contrast to IVTT-based microarrays, printing purified protein allows a highly quantifiable approach to be taken. Affinity purification and dialysis of expression products substantially reduces the risk of background reactivity to bacterial components, and the simple determination of target protein concentrations allows defined quantities of product to be spotted, providing much greater confidence when comparing reactivity between targets. However, these advantages come at a substantial cost; the need for in silico analysis to design vectors, transfection procedures, expression and purification drastically slows the rate at which putative targets can be produced and screened. Shorter, epitope specific sequences may in theory

be transposed from IVTT systems with a view to generating more granular serological screens, though we accept that truncated protein targets will in some cases favour linear B cell epitopes, while missing conformational epitopes. However, for measuring exposure to infection there is less importance on the targeting of confirmation epitopes than would be required for protective epitopes [57].

The primary benefit of the microarray platform is the ability to screen orders of magnitude more targets simultaneously than more standard serological assays. Our analysis shows that both IVTT and purified proteins can be successfully used to capture malarial protein-antigen specific antibody responses on a protein microarray. Although correlations of response between expression systems are not as strong as may have been expected, a number of acknowledged technical differences in the methods of protein production may account for this finding. In addition to the *E. coli* in vivo and IVTT systems utilised here, high-throughput wheat germ cell free systems have been successfully used to conduct large scale serological screens of putative antigen biomarkers [74, 75], alongside chemically synthesised peptide arrays [57, 62]. High-throughput mammalian and baculovirus expression systems have also been pioneered for the production of recombinant proteins [36, 76]. Differences in expression efficiency and the homology to native epitopes achieved by the assortment of available approaches likely have considerable impact on the capture of antibody from sample. This variability should be accounted for both in terms of choosing an experimental approach and comparative analysis between different methods. We suggest that further investigation of differences in seroreactivity to sequence-matched proteins derived from contrasting expression systems is needed to shed light on the parity between such data that is already widely published. It should also be noted that it is unlikely that any single expression platform will satisfy the demands of all recombinant expression projects due to varying importance such as protein folding, proteins activity (e.g. enzymes) and glycosylation. In addition, *E. coli* expression has the advantage of low cost, flexibility and easy scale-up.

Considering the data presented here more broadly, observed trends lend support to the utilisation of both IVTT and purified arrays depending on the objectives and context of hypotheses to be investigated. The strengths and weaknesses of each expression system should dictate the chosen approach on a case-by-case basis. For example, very high-density proteome level screening to identify 'shortlists' of candidate markers based on binary categorisation of seropositivity may be best achieved using IVTT systems. In contrast, smaller numbers of 'shortlisted' targets expressed as purified proteins may allow for more nuanced characterisation of antibody responses on a more continuous scale. As already described, the key limitation in the production of purified recombinants in our current expression pipeline is throughput. The adaption of our methods to increase the capacity of protein production would improve our ability to more widely mine the biomarker information derived from the IVTT platform. As such, we are currently exploring a number of existing approaches to address this methodological bottleneck [38, 77].

In summary, the IVTT protein microarray approach has proven to be a powerful, high-throughput, biomarker discovery platform with applicability across a range of infectious diseases. When combined with a cheap, scalable and flexible protein expression platform such as the *E. coli* in vivo expression platform we have the ability to mine potential diagnostic and vaccine related targets.

## Supporting information

**S1 Fig. Data normalisation processes for IVTT and purified protein spots.** After local background correction using the backgroundCorrect function from the limma package, purified

protein spots were additionally corrected for possible GST reactivity by subtracting GST reactivity using the same function. After Log2 transformation, IVTT and purified proteins were normalised to background control spots of empty T7 vector and PBS buffer control spots respectively.
(PDF)

**S2 Fig.** Correlogram of multiple antigen-matched targets (left). Spearman's rank correlation reported (rs) and increasing blue colour scale indicates relative strength of correlation based on calculated correlations for all proteins included in this analysis. Protein schematic (right) represents amino-acid aligned representation of IVTT (green) and purified (orange) proteins to the full-length native protein (grey). Proteins in the correlogram and schematic are correspondingly aligned.
(PDF)

**S3 Fig. Magnitude and range of response to IVTT and purified proteins, stratified by age.** All sample responses (n = 899) to all protein targets grouped by antigen, presented with median and interquartile range.
(PDF)

**S1 Table. Detail of expressed protein targets.** A key to the simplified nomenclature used for specific proteins in text is provided.
(XLSX)

**S2 Table. Correlation coefficient results for all protein pairs.** Protein targets are grouped by antigen, and all possible combinations within each antigen group are shown.
(XLSX)

## Acknowledgments

We thank all the study participants who participated in the original PRISM study (Program for Resistance, Immunology, Surveillance, and Modelling of Malaria in Uganda; East Africa ICEMR) from which the test samples were taken. The authors give thanks to James Beeson for the provision of purified recombinant protein EBA140 RIII-V, Ross Coppel for the provision of MSP4 and MSP5, and Tony Holder for the provision of GAMA and MSP7.

## Author Contributions

**Conceptualization:** Isabel Rodriguez, Bryan Greenhouse, Chris Drakeley, Phil L. Felgner, Kevin K. A. Tetteh.

**Data curation:** Tate Oulton, Joshua Obiero.

**Formal analysis:** Tate Oulton, Joshua Obiero, Will Stone, Kevin K. A. Tetteh.

**Funding acquisition:** Chris Drakeley, Kevin K. A. Tetteh.

**Investigation:** Tate Oulton, Joshua Obiero, Isabel Rodriguez, Bryan Greenhouse, Chris Drakeley, Will Stone, Kevin K. A. Tetteh.

**Methodology:** Tate Oulton, Joshua Obiero, Rebecca A. Dabbs, Phil L. Felgner, Will Stone.

**Project administration:** Christine M. Bachman.

**Resources:** Isaac Ssewanyana, Rebecca A. Dabbs, Christine M. Bachman, Phil L. Felgner, Kevin K. A. Tetteh.

**Supervision:** Will Stone, Kevin K. A. Tetteh.

**Visualization:** Tate Oulton, Joshua Obiero.

**Writing – original draft:** Tate Oulton, Kevin K. A. Tetteh.

**Writing – review & editing:** Tate Oulton, Joshua Obiero, Isabel Rodriguez, Isaac Ssewanyana, Rebecca A. Dabbs, Christine M. Bachman, Bryan Greenhouse, Chris Drakeley, Phil L. Felgner, Will Stone, Kevin K. A. Tetteh.

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
