## [Decision Letter · Decision Letter 0]

22 Dec 2021

PONE-D-21-37402Plasmodium falciparum serology: A comparison of two protein production methods for analysis of antibody responses by protein microarrayPLOS ONE

Dear Dr. Tetteh,

Thank you for submitting your manuscript to PLOS ONE. After careful consideration, we feel that it has merit but does not fully meet PLOS ONE’s publication criteria as it currently stands. Therefore, we invite you to submit a revised version of the manuscript that addresses the points raised during the review process.

Please take all the comments provided by two reviewers who are experts in this specific research field into consideration in your revision. And please also provide point-by-point responses. 

We look forward to receiving your revised manuscript.

Kind regards,

Takafumi Tsuboi

Academic Editor

PLOS ONE

Journal Requirements:

"We thank all the study participants who participated in the original PRISM study (Program for Resistance, Immunology, Surveillance, and Modelling of Malaria in Uganda; East Africa ICEMR) from which the test samples were taken. The authors give thanks to James Beeson for the provision of purified recombinant protein EBA140 RIII-V, Ross Coppel for the provision of MSP4 and MSP5, and Tony Holder for the provision of GAMA and MSP7. Kevin K.A.Tetteh was supported by a Bloomsbury SET Award (Innovation Fellowship to KKAT; BSA14) under the UKRI Connecting Capabilities Fund (CCF)."

"This work was supported by funding from the Global Good Fund I, LLC. The funders had no role in study design, data collection and analysis, decision to publish, or preparation of the manuscript."

Reviewers' comments:

Reviewer's Responses to Questions

**Comments to the Author**

1. Is the manuscript technically sound, and do the data support the conclusions?

Reviewer #1: Yes

Reviewer #2: Yes

2. Has the statistical analysis been performed appropriately and rigorously? 

Reviewer #1: Yes

Reviewer #2: Yes

3. Have the authors made all data underlying the findings in their manuscript fully available?

Reviewer #1: No

Reviewer #2: Yes

4. Is the manuscript presented in an intelligible fashion and written in standard English?

Reviewer #1: Yes

Reviewer #2: Yes

5. Review Comments to the Author

Reviewer #1: In this study, Oulton et al compared the reactivity of recombinant proteins synthesized using different platforms. Specifically, they compared human sera reactivity to purified or unpurified recombinant proteins synthesized in E . coli /Pichia pastoris or an in vitro transcription/ translation (IVTT) platform, respectively.

Although the study is important since it attempts to address an existing gap, it has several weaknesses. First, it fails to acknowledge the great progress in antigen discovery that have been made in the last decade rather basing the current work on studies conducted more than a decade. The authors only briefly mention the high-throughput studies in the discussion. The introduction section should updated to clearly capture the current status as reported in more recent references. Second, key data is missing or the design didn’t not fully answer the researchers questions. The purity of the proteins is unknown since no data was presented. Moreover, it’s impossible to tell whether the low signals observed with the IVTT expressed proteins was due to failure of expression or was due to poor quality of the proteins. Since the IVTT expression system is scalable, at least several purified IVTT proteins should have been included as controls.

Specific comments

Line 46-47 is not clear what “the rate of IVTT” means. This line needs editing.

For clarity, the authors should – at first appearance- clearly state that they compared the reactivity of recombinant proteins expressed in E. coli or E. coli based IVTT proteins expression system. As it is now, it not clear how the “purified proteins” were synthesized.

Since this study is about assessing the effect of purification on proteins, the SDS-PAGE images, of the study proteins should be made available to the readers.

Line 110-112: “Purified protein targets were smaller than their IVTT counterparts; purified protein targets are produced as fragments of the full-length protein, with the aim of capturing antibodies specific to predicted epitopes based on in silico analysis”. Does it mean that the IVTT proteins were not detecting antibodies specific to predicted epitopes? Please correct.

Did the empty vectors express GST or the tags was expressed separately for background GST fluorescence correction? How were the His-tagged proteins normalized?

Line 188-190: If the amino acid labeling is correct, the difference was only at the the c-terminal amino acid but not N-terminal.

I note that this study was part of a very comprehensive longitudinal study in Uganda. Can the authors confirm that only one Ugandan researcher should be included as a co-authors?

Reviewer #2: The study by Oulton, et al, describes the ability of IgG antibodies to bind to Pf antigens whether produced by an IVTT system or through recombinant whole-cell system (and purified). This is an important question for the malaria serological community, as high-throughput fishing expeditions can lead to downstream decision making, but results are rarely compared to the selected produced product. The data shows, not surprisingly, that the recombinant antigens have a better capacity for IgG capture compared to comparable targets produced by IVTT.

Major comments

A major criticism of this work is how the authors compare the ability of IgG capture only through the microarray platform on nitrocellulose slides. The reviewer would be certain that these samples (or a subset) have also been assayed by a separate immunoassay platform such as ELISA or multiplex bead assay. Comparison of signal intensity for single persons’ samples between the microarray and one of these standard IgG detection assays would be an important bridge for the laboratorian to translate these findings.

The concept of background signal due to a crude IVTT system with a lot of E. coli protein in it is brought up a few times, though no data is shown for what this background was in the authors’ hands. Was the background MFI of the IVTT on the microarray considerably higher than for the purified antigens? Did this lead to an overall reduced signal IVTT targets after normalization?

Minor comments

Line 25-26: “lack of a clearly-defined relationship between…” confusing text here, should consider re-wording

Line 46-47: “rates of IVTT..” rates meaning hours expended, time for each target, other? Please provide a little more detail here regarding ‘rates’

Line 49: write out E. coli first time used

Line 53: Sentence starting with ‘However’ can be removed

Line 63: say why the production of complex conformational proteins “can sometimes be a challenge”

Table 1: “Description of P. falciparum antigens and…”

Figure 1: either in this panel (or a new one) it would actually be more informative for the reader to have a line connecting the IVTT MFI value to the ‘purified’ value for each individual target. In that way for a single target, the relative log increase in signal is clearly shown.

Line 214-217: the authors have argued previously that IVTT is more appropriate for the broad screening of biomarker identification. In fact, a recombinant and purified antigen is no longer a ‘candidate’, but there would have been a reason for going to all the trouble to create it in a cell system.

Throughout: the authors should note that the lack of correlation for many of their targets is due to the fact that the purified antigen is maxing out the MFI signal whereas is seems IVTT targets never do this.

6. PLOS authors have the option to publish the peer review history of their article (what does this mean?). If published, this will include your full peer review and any attached files.

Reviewer #1: No

Reviewer #2: No

---

## [Author Response · Author response to Decision Letter 0]

19 Jul 2022

Reviewer #1: In this study, Oulton et al compared the reactivity of recombinant proteins synthesized using different platforms. Specifically, they compared human sera reactivity to purified or unpurified recombinant proteins synthesized in E . coli /Pichia pastoris or an in vitro transcription/ translation (IVTT) platform, respectively.

Although the study is important since it attempts to address an existing gap, it has several weaknesses. First, it fails to acknowledge the great progress in antigen discovery that have been made in the last decade rather basing the current work on studies conducted more than a decade. The authors only briefly mention the high-throughput studies in the discussion. The introduction section should updated to clearly capture the current status as reported in more recent references.

Author response:

We thank the reviewer for this comment. This omission was not intentional. Additional text and supporting references have been included into the introduction to address this gap as follows:

Line 46-57

“Cell-free synthesis (CFS) is a technique first established over 50 years ago as a means to dissect the molecular mechanisms around protein expression. More recently, the technique has been used as a high throughput expression platform to explore a number of diverse biological processes (Dopp et al Synthesis and systems biology 4, PMID: 31750411; Harini et al. Current Opinions in Biotechnology 2022, PMID: 35247659). At its simplest, the approach utilises the crude extract containing the transcription and translation machinery from the cell, performing the process of protein expression without the constraints of the cell. This allows a wide variety of proteins to be expressed including those that would be deemed toxic if expression was attempted within the confines of the cell membrane (Silverman et al Nat Rev Gen 2019, PMID: 31780816). CFS systems based on Escherichia coli (E.coli) are among the most widely used of the IVTT systems (Harini et al Current Opinions in Biotechnology 2022, PMID: 35247659) and have helped to transform the narrative around a number of areas including biomarker discovery for infectious diseases (Davies DH et al. Proc Natl Acad Sci U S A. 2005 Jan 18;102(3):547-52. PMID: 15647345; Venkatesh A et al. Methods Mol Biol. 2021;2344:139-150. PMID: 34115357; Kobayashi T et al. mSphere. 2019 Mar 27;4(2):e00061-19. PMID: 30918058). Despite the widespread uptake of the approach there remain some issues with the technique. This includes significant heterogeneity of expression, leading some research groups to describe the mechanisms of the process as a “black box”.”

 Second, key data is missing or the design didn’t not fully answer the researchers questions. The purity of the proteins is unknown since no data was presented. Moreover, it’s impossible to tell whether the low signals observed with the IVTT expressed proteins was due to failure of expression or was due to poor quality of the proteins. Since the IVTT expression system is scalable, at least several purified IVTT proteins should have been included as controls.

Author response:

We thank the reviewer for their comments. The data is not missing but is in fact a recognised fundamental weakness in the IVTT platform. The low signals described in this study and covered extensively in the literature is a known limitation of the system. The high throughput nature of the system is somewhat tempered by the high level of heterogeneity in protein production that is inherent with the system. The IVTT microarray approach is more of an ‘opt in’ where positives can be further validated using other approaches. This circumvents the heterogeneity which can result in detectable signal in one expression run vs no signal in another. We have included additional text highlighting the state-of-the-art with regards to the IVTT platform including the following references in the introduction (Dopp et al Synthesis and systems biology 4, PMID: 31750411; Harini et al. Current Opinions in Biotechnology 2022, PMID: 35247659). 

Although the system is scalable, in terms of volume of reaction mix/culture, this does not circumvent the known heterogeneity that is inherent within the system. This is an issue that groups are trying to address as an end goal. Large-scale systems do exist within the biotechnology industry, but these are not generally accessible to researchers due to space considerations and cost to run (Silverman et al Nat Rev Gen 2019, PMID: 31780816)

Relevant controls are built into the array and are described under heading “Protein microarray”. 

Refer to inserted text as above (Line 46-57)

Specific comments

Line 46-47 is not clear what “the rate of IVTT” means. This line needs editing.

For clarity, the authors should – at first appearance- clearly state that they compared the reactivity of recombinant proteins expressed in E. coli or E. coli based IVTT proteins expression system. As it is now, it not clear how the “purified proteins” were synthesized.

Since this study is about assessing the effect of purification on proteins, the SDS-PAGE images, of the study proteins should be made available to the readers.

Author response:

The line previously at 46-47 has been deleted and replaced with more detailed text as highlighted above for Line 46-57

We agree with the reviewer regarding the difference in expression systems and have reworked the text for clarity as follows:

For clarity proteins produced using the IVTT system will simply be referred to as IVTT proteins, and those produced by conventional E.coli expression will be referred to as purified proteins. The text has been edited as follows:

Line 77-79

“For clarity proteins produced using the IVTT system will simply be referred to as IVTT proteins, and those produced by conventional E.coli expression will be referred to as purified proteins.”

The purified proteins described here have been fully described elsewhere and so were not duplicated here. 

Line 110-112: “Purified protein targets were smaller than their IVTT counterparts; purified protein targets are produced as fragments of the full-length protein, with the aim of capturing antibodies specific to predicted epitopes based on in silico analysis”. Does it mean that the IVTT proteins were not detecting antibodies specific to predicted epitopes? Please correct.

Did the empty vectors express GST or the tags was expressed separately for background GST fluorescence correction? How were the His-tagged proteins normalized?

Author response:

This section has now been clarified as follows:

Line 125-1323

“Sequence information used in the design and expression of the purified E.coli proteins were generally smaller than the equivalent proteins expressed in the IVTT cell-free systems. This was done to limit the sequence length to below 1kb as expression of proteins larger that 1kb in E.coli can contribute to poor or failed expression yields (Vedadi M et al. Mol Biochem Parasitol. 2007 Jan;151(1):100-10. PMID: 17125854; Mehlin C et al. Mol Biochem Parasitol. 2006 Aug;148(2):144-60. PMID: 16644028). Truncation of target sequences was based on in silico mapping of each protein sequence to focus on regions of predicted immunogenicity based on the in silico analysis. Empty GST vectors were expressed and the purified GST used in background correction for proteins with this tag. His-tag vector was not expressed as it has proven impossible to express and purify the 6xhistidine tag in isolation.”

Line 188-190: If the amino acid labeling is correct, the difference was only at the the c-terminal amino acid but not N-terminal.

Author response:

As part of the design of the protein targets for expression, the signal peptide and any transmembrane proteins are removed from the sequences, which is standard practice with expression of target proteins in E.coli. Meaning that some edits can be based around the signal peptide only, resulting in minor truncations of the target sequences. Inclusion of the signal peptide and transmembrane domains are key factors in the poor or failed expression of recombinant proteins. 

Supplementary Table S1 provides detailed information on each target to help illustrate this. 

I note that this study was part of a very comprehensive longitudinal study in Uganda. Can the authors confirm that only one Ugandan researcher should be included as a co-authors?

Author response:

We thank the reviewer for this comment. As the lead of our Equity and Diversity committee and one of the key drivers in pushing for better equity in research I am encouraged by this question and thank the reviewer for raising this. The inclusion of authors into this manuscript was conducted in an open and fair manner, with final decision for author inclusion from Uganda resting with Dr Ssewanyana (Director of Laboratory Services Uganda National Health Laboratory Services). Ultimately, the inclusion of Ugandan staff relevant to the project rested with him, and this is a decision all authors supported. In addition, there have been over 246 publications from PRISM 1 and 2 for which there has been equitable contribution in terms of authorship across the collaborative partners (https://www.niaid.nih.gov/research/east-africa-international-center-excellence-malaria-research).

Reviewer #2: The study by Oulton, et al, describes the ability of IgG antibodies to bind to Pf antigens whether produced by an IVTT system or through recombinant whole-cell system (and purified). This is an important question for the malaria serological community, as high-throughput fishing expeditions can lead to downstream decision making, but results are rarely compared to the selected produced product. The data shows, not surprisingly, that the recombinant antigens have a better capacity for IgG capture compared to comparable targets produced by IVTT.

Major comments

A major criticism of this work is how the authors compare the ability of IgG capture only through the microarray platform on nitrocellulose slides. The reviewer would be certain that these samples (or a subset) have also been assayed by a separate immunoassay platform such as ELISA or multiplex bead assay. Comparison of signal intensity for single persons’ samples between the microarray and one of these standard IgG detection assays would be an important bridge for the laboratorian to translate these findings.

The concept of background signal due to a crude IVTT system with a lot of E. coli protein in it is brought up a few times, though no data is shown for what this background was in the authors’ hands. Was the background MFI of the IVTT on the microarray considerably higher than for the purified antigens? Did this lead to an overall reduced signal IVTT targets after normalization?

Author response:

The comparison of IgG in this study using the protein microarray multiplex assay was performed based on prior research from a number of groups including ours using the monoplex ELISA to assess antibody responses to infection based primarily on IgG but also targeting subclass and isotype responses (van den Hoogen LL et al. Front Microbiol. 2019 Jan 16;9:3300; Kamuyu G et al. Front Immunol. 2018 Dec 11;9:2866. PMID: 30619257; King CL et al. Am J Trop Med Hyg. 2015 Sep;93(3 Suppl):16-27. PMID: 26259938; Helb DA et al. Proc Natl Acad Sci U S A. 2015 Aug 11;112(32):E4438-47. PMID: 26216993). Many of the key markers of seroincidence have also been extensively assessed using the Luminex multiplex bead-based array (Wu L et al. EClinicalMedicine. 2022 Feb 14;44:101272. PMID: 35198913; Achan J et al. Lancet Microbe. 2022 Jan;3(1):e62-e71. PMID: 34723228). Comparisons between the microarray and ELISA have already been (Ondigo BN et al. Malar J. 2012 Dec 21;11:427. PMID: 23259607) and so was not repeated here. 

Primarily, the IVTT is a uniquely high-throughput platform capable of generating 100’s to 1000’s of targets in parallel. However, levels of expression can be highly variable. Our study was intended to compare the E.coli IVTT vs the E.coli purified protein responses, specifically within the context of microarray analysis. This study was intended to support the assertion that biomarker discovery can feed into experimental pipelines that can be interrogated using high resolution methods. 

Data processing for the IVTT microarray has been presented extensively elsewhere. No. Background MFI for the IVTT did not show higher background responses that their purified antigen counterparts. As such there was not bias in terms of an overall reduced signal for the IVTT proteins. Differences in signal intensity were due to the inherent heterogeneity of expression with the IVTT platform as described earlier. 

Minor comments

Line 25-26: “lack of a clearly-defined relationship between…” confusing text here, should consider re-wording.

Author response:

We agree with the reviewer and have edited the sentence for clarity from:

“Despite the lack of a clearly defined relationship between antigen-matched targets produced in each expression system, our data indicate that protein microarrays produced using either method can be used confidently, in a context dependent manner, though care should be taken when comparing data derived from contrasting approaches.”

To, 

Line 25-29

“Despite the lack of a clear correlation between antigen-matched recombinant proteins from each expression system, our data indicates that protein microarrays produced using either method can be used confidently, in a context dependent manner, though care should be taken when comparing data derived from contrasting approaches.”

Line 46-47: “rates of IVTT..” rates meaning hours expended, time for each target, other? Please provide a little more detail here regarding ‘rates’

Author response:

This section has been reworked for improved clarity as follows:

The line at 46-47 has been deleted and replaced with more detailed text as follows:

Line 46-57

“Cell-free synthesis (CFS) is a technique first established over 50 years ago as a means to dissect the molecular mechanisms around protein expression. More recently, the technique has been used as a high throughput expression platform to explore a number of diverse biological processes (Dopp et al Synthesis and systems biology 4, PMID: 31750411; Harini et al. Current Opinions in Biotechnology 2022, PMID: 35247659). At its simplest, the approach utilises the crude extract containing the transcription and translation machinery from the cell, performing the process of protein expression without the constraints of the cell. This allows a wide variety of proteins to be expressed including those that would be deemed toxic if expression was attempted within the confines of the cell membrane (Silverman et al Nat Rev Gen 2019, PMID: 31780816). CFS systems based on Escherichia coli (E.coli) are among the most widely used of the IVTT systems (Harini et al Current Opinions in Biotechnology 2022, PMID: 35247659) and have helped to transform the narrative around a number of areas including biomarker discovery for infectious diseases (Davies DH et al. Proc Natl Acad Sci U S A. 2005 Jan 18;102(3):547-52. PMID: 15647345; Venkatesh A et al. Methods Mol Biol. 2021;2344:139-150. PMID: 34115357; Kobayashi T et al. mSphere. 2019 Mar 27;4(2):e00061-19. PMID: 30918058). Despite the widespread uptake of the approach there remain some issues around the technique. This includes significant heterogeneity of expression, leading some research groups to describe the mechanisms of the process as a “black box”. “ 

Line 49: write out E. coli first time used

Author response:

Line 53

We have made this change.

Line 53: Sentence starting with ‘However’ can be removed

Author response:

Line 63

We have removed, ”However”.

Line 63: say why the production of complex conformational proteins “can sometimes be a challenge”

Author response:

Expression of recombinant proteins is challenging. This is in part due to the expression of proteins foreign to the bacteria, the speed at which bacteria express proteins, only partially mitigated with a reduction in expression temperature; and the lack of essential molecular chaperones to aid correct folding/refolding of proteins (Francis and Page Curr Protoc Protein Sci. 2010 Aug; 61(1): 5241–52429.)

The following sentence has been included for clarity:

Line 73-76

“These challenges are in part due to the expression of proteins foreign to the bacteria, the speed at which bacteria express proteins, only partially mitigated with a reduction in expression temperature; and the lack of essential molecular chaperones to aid correct folding/refolding of proteins (Francis and Page Curr Protoc Protein Sci. 2010 Aug; 61(1): 5241–52429; Vedadi M et al. Mol Biochem Parasitol. 2007 Jan;151(1):100-10. PMID: 17125854; Mehlin C et al. Mol Biochem Parasitol. 2006 Aug;148(2):144-60. PMID: 16644028).

Table 1: “Description of P. falciparum antigens and…”

Author response:

The recommended edit has been made

Figure 1: either in this panel (or a new one) it would actually be more informative for the reader to have a line connecting the IVTT MFI value to the ‘purified’ value for each individual target. In that way for a single target, the relative log increase in signal is clearly shown.

Author response:

We appreciate the reviewer’s comments. However, the recommended edits would lead to a less transparent figure. In figure 1, we aim to demonstrate differences in reactivity between proteins produced by the two expression systems overall, rather than between specific antigens. Antigens were colour coded to allow for a basic comparison of reactivity between antigen matched targets, though this is not the main purpose of this figure. We would instead like to draw the reviewer’s attention to Figure 2, which highlights both the magnitude and range of responses of the IVTT proteins compared to the purified proteins. 

Similarly, to Figure 3 and supplementary Figure 2 which show the correlation of responses between the IVTT and the purified proteins. 

We believe that these figures capture what the reviewer was trying to highlight.

Line 214-217: the authors have argued previously that IVTT is more appropriate for the broad screening of biomarker identification. In fact, a recombinant and purified antigen is no longer a ‘candidate’, but there would have been a reason for going to all the trouble to create it in a cell system.

Throughout: the authors should note that the lack of correlation for many of their targets is due to the fact that the purified antigen is maxing out the MFI signal whereas is seems IVTT targets never do this.

Author response:

We thank the reviewer for their comments and have included additional text highlighting the state-of-the-art with regards to the IVTT platform. The low signals described in this study and covered extensively in the literature is a known limitation of the system. The high throughput nature of the system is somewhat tempered by the high level of heterogeneity in protein production that is inherent with the system. The IVTT microarray approach is more of an ‘opt in’ where positives can be further validated using other approaches. This circumvents the heterogeneity which can result in detectable signal in one expression run vs no signal in another. This is fully covered in the following references and has been included in the introduction (Dopp et al Synthesis and systems biology 4, PMID: 31750411; Harini et al. Current Opinions in Biotechnology 2022, PMID: 35247659). 

Although the system is scalable, in terms of volume of reaction mix/culture, this does not circumvent the known heterogeneity that is inherent within the system. This is an currently an area of active research. Large-scale systems do exist within the biotechnology industry, but these are not generally accessible to researchers due to space considerations and cost to run (Silverman et al Nat Rev Gen 2019, PMID: 31780816).

Refer to Line 46-57

Relevant controls are built into the array and are described under the heading “Protein microarray”.

---

## [Decision Letter · Decision Letter 1]

3 Aug 2022

Plasmodium falciparum serology: A comparison of two protein production methods for analysis of antibody responses by protein microarray

PONE-D-21-37402R1

Dear Dr. Tetteh,

We’re pleased to inform you that your manuscript has been judged scientifically suitable for publication and will be formally accepted for publication once it meets all outstanding technical requirements.

Kind regards,

Takafumi Tsuboi

Academic Editor

PLOS ONE

Additional Editor Comments (optional):

Reviewers' comments:

Reviewer's Responses to Questions

**Comments to the Author**

1. If the authors have adequately addressed your comments raised in a previous round of review and you feel that this manuscript is now acceptable for publication, you may indicate that here to bypass the “Comments to the Author” section, enter your conflict of interest statement in the “Confidential to Editor” section, and submit your "Accept" recommendation.

Reviewer #1: All comments have been addressed

Reviewer #2: All comments have been addressed

2. Is the manuscript technically sound, and do the data support the conclusions?

Reviewer #1: Yes

Reviewer #2: Yes

3. Has the statistical analysis been performed appropriately and rigorously? 

Reviewer #1: Yes

Reviewer #2: Yes

4. Have the authors made all data underlying the findings in their manuscript fully available?

Reviewer #1: (No Response)

Reviewer #2: Yes

5. Is the manuscript presented in an intelligible fashion and written in standard English?

Reviewer #1: (No Response)

Reviewer #2: Yes

6. Review Comments to the Author

Reviewer #1: (No Response)

Reviewer #2: (No Response)

7. PLOS authors have the option to publish the peer review history of their article (what does this mean?). If published, this will include your full peer review and any attached files.

Reviewer #1: No

Reviewer #2: No

---

## [Editor Report · Acceptance letter]

19 Aug 2022

PONE-D-21-37402R1 

*Plasmodium falciparum* serology: A comparison of two protein production methods for analysis of antibody responses by protein microarray 

Dear Dr. Tetteh:

I'm pleased to inform you that your manuscript has been deemed suitable for publication in PLOS ONE. Congratulations! Your manuscript is now with our production department. 

Kind regards, 

on behalf of

Prof. Takafumi Tsuboi 

Academic Editor

PLOS ONE